# LDCNet: Long-Distance Context Modeling for Large-Scale 3D Point Cloud Scene Semantic Segmentation

## ABSTRACT

Large-scale point cloud semantic segmentation is a challenging task in 3D computer vision. A key challenge is how to resolve ambiguities arising from locally high inter-class similarity. In this study, we introduce a solution by modeling long-distance contextual information to understand the scene's overall layout. The context sensitivity of previous methods is typically constrained to small blocks(e.g. $2m \times 2m$) and cannot be directly extended to the entire scene. For this reason, we propose **L**ong-**D**istance **C**ontext Modeling Network(LDCNet). Our key insight is that keypoints are enough for inferring the layout of a scene. Therefore, we represent the entire scene using keypoints along with local descriptors and model long-distance context on these keypoints. Finally, we propagate the long-distance context information from keypoints back to non-keypoints. This allows our method to model long-distance context effectively. We conducted experiments on six datasets, demonstrating that our approach can effectively mitigate ambiguities. Our method performs well on large, irregular objects and exhibits good generalization for typical scenarios.

## CCS CONCEPTS

• **Computing methodologies → Scene understanding**.

## KEYWORDS

Long Distance Context, Key Points, Large-Scale Scene

## 1 INTRODUCTION

Point cloud semantic segmentation aims at predicting point-wise categories, which has an important role in the fields of autonomous driving, augmented reality, digital preservation, and robotics. In recent years, many deep networks for point cloud semantic segmentation have been proposed[4, 8, 9, 14, 15, 18, 20, 24, 26, 29, 30, 35, 41, 42, 45, 51, 52, 54]. These methods achieve appealing results. However, they mostly focus on object-level point clouds, and few designs and experiments have been conducted on scene-level point clouds. Based on the fact that point clouds are disordered and unstructured, scene-level point cloud semantic segmentation is still a challenging task.

A key challenge is addressing the issue of ambiguity resulting from local inter-class similarity. For instance, in certain indoor

Permission to make digital or hard copies of all or part of this work for personal or classroom use is granted without fee provided that copies are not made or distributed for profit or commercial advantage and that copies bear this notice and the full citation on the first page. Copyrights for components of this work owned by others than the author(s) must be honored. Abstracting with credit is permitted. To copy otherwise, or republish, to post on servers or to redistribute to lists, requires prior specific permission and/or a fee. Request permissions from permissions@acm.org.
*ACM MM, 2024, Melbourne, Australia*
© 2024 Copyright held by the owner/author(s). Publication rights licensed to ACM.
ACM ISBN 978-x-xxxx-xxxx-x/YY/MM
https://doi.org/10.1145/nnnnnnn.nnnnnnn

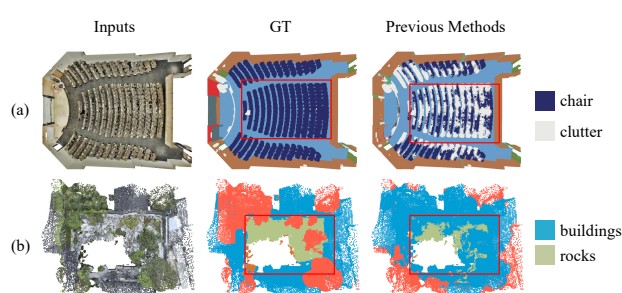

**Figure 1: Example large-scale scenes with irregular large-sized objects. (a) auditorium room from the S3DIS dataset. (b) rockery garden from the CHNRockery3D dataset. Previous methods such as SPG[19], RandLANet[14] and PointTr[52] give similar results as shown in the 3rd column.**

scenes, the network can effectively segment individual chairs almost without ambiguity. However, when multiple chairs are placed together, the network tends to classify them as clutter(Figure 1(a)). Similarly, In outdoor scenes, such as Chinese classical gardens featuring rockeries and ponds made of stones, the overall look of a pond may resemble a local region of a rockery. This similarity poses a challenge for the network in distinguishing between the two (Figure 1(b)). Additionally, some parts of buildings exhibit high geometric similarity with local areas of rockeries, leading to confusion. This ambiguity hinders the network's ability to effectively handle such scenarios.

Our observation is that modeling long-distance contexts to capture the spatial layout of the entire scene and the relation between objects can effectively mitigate the issue of ambiguity. For instance, within a classroom setting, chairs are typically positioned at the center of the scene. Additionally, in garden scenarios, ponds are usually positioned at lower elevations, while buildings generally surround the entire scene. Furthermore, rockeries are typically situated near the buildings.

To model long-distance context, early methods[11, 19] segmented the scene into blocks and employed Recurrent Neural Networks (RNNs) to capture contextual information between these blocks. However, due to the limited long-term modeling capacity of RNNs, extending the context to cover the entire scene was challenging. Inspired by the impressive long-distance modeling capabilities of Transformers in 2D image and NLP domains[6, 10, 25, 27, 37, 48], many methods explored using Transformers for capturing long-distance context in point clouds. However, as the memory footprint of Transformers grows quadratically with the number of points, existing methods typically fall into two categories. One confines the input point cloud to a small block (e.g., 2m x 2m) and captures global context on a sparse subset of points (usually 1024 or 2048 points) within these small blocks[23, 50]. The other attempts to

use a larger number of points (e.g., 100,000) [17, 18, 52]. To address the cost issue, these methods attempt to use Transformers locally (e.g., 16 points in k-neighbor or 3d window), with the long-distance context being implicitly acquired during the network's forward propagation. These methods either restrict the context range to small blocks or require very deep networks to extend the context to cover the entire scene.

To this end, we propose **L**ong-**D**istance **C**ontext Network(LDCNet). Our idea is simple and intuitive, the key points are sufficient to reason about the layout of the scene. Hence, we transform large-scale scenes into a representation based on key points and model long-distance context on these key points. We then aggregate contextual information from key points into non-key points for dense prediction. With the number of key points significantly fewer than the original point cloud, it's feasible to model long-distance context across the entire scene. Experiments demonstrate that our method extends the context length beyond 10 meters without adding too many network parameters and memory footprint. Our network accurately perceives the full view of irregular large-sized objects and provides correct predictions.

Our main contributions are as follows

- We address the problem of high inter-class similarity in semantic segmentation of large-scale scene point clouds. We introduce a new Transformer-based method that efficiently models the context of the entire scene and reasons about the overall scene layout relationships.
- We model the long-distance context by first extracting key points from a point cloud and carefully initiating a descriptor for each key point. Then we apply Transformer globally on all key points to extract long-distance contextual information. Finally, we aggregate this information into non-key points for dense prediction.
- Experiments show our approach can effectively mitigate ambiguities by expanding the contextual range to the entire scene (over 10m). Our method performs well on large, irregular objects in large-scale scenes, and it also exhibits good generalization for typical scenarios.

## 2 RELATED WORK

**Point cloud semantic segmentation.** Unlike grid-like 2D image data, the disordered and unstructured nature of point clouds poses challenges and difficulties in the design of point cloud deep networks. PointNet[29] was the first to explore the design paradigm of point cloud deep networks and to address the problem of permutation invariance. Many subsequent approaches have followed this idea[9, 12, 18, 20, 30, 35, 41, 42, 51, 52]. Though these methods have achieved appealing results on object-level point cloud semantic segmentation, the large-scale scene semantic segmentation is still under-explored.

**Context modeling for point clouds.** The methods can be broadly categorized into three groups. One group restricts the input point cloud to small blocks, allowing the standard self-attention mechanism to directly extract global context due to the small spatial extent and point count within each block[23, 50]. However, this approach is limited to small chunks, rendering it ineffective for large blocks (i.e. the entire scene) with a high number of points.

In response, another group of methods attempts to enhance the local receptive field at each layer, implicitly extending the range of contexts modeled as the network deepens[17, 18, 24, 39, 52]. Our experiments reveal that achieving an effective context for the entire scene requires considerable network depth using this approach. The final group of methods aims to convert a 3D representation into a 2D representation; for instance, RangeFormer[16] transforms a point cloud into a range image, extracting long-distance context in 2D. However, this method is best suited for sparse point clouds, as dense point clouds risk losing significant information. Compared to these methods, in terms of context sensitivity, our method can cover long-distance ranges or even the entire scene. For representation, we utilize key points and local descriptors, effectively preserving 3D information.

**Large-scale point cloud scene semantic segmentation.** Few methods have been specifically tailored for large-scale scenes. SP-Graph [19] preprocesses extensive point clouds into super-point graphs to learn per-super-point semantics. They employ an RNN to capture contextual information between superpoints. However, the limited long-term modeling capability of RNNs hinders the coverage of context over great distances. RandLA-Net[14] proposes using random sampling instead of farthest sampling to expedite network inference. They employ the concept of 'dilation'[47] to locally expand the receptive field, aiming to continuously broaden the context during network feedforward. Nonetheless, achieving context extension to the entire scene requires a very deep network.

**Vision Transformer.** In recent years, the Transformer model has achieved remarkable success in modeling powerful long-distance context in NLP and 2D image domains[10, 25, 27, 37, 38, 48]. While Transformers are widely popular in 2D and NLP, their application to point clouds has been relatively unexplored until recently. Several transformer-based point cloud networks have emerged[18, 28, 52]. For instance, PointTr[52] applies the Transformer in a local k-neighborhood, while StratifiedTr[18] applies the transformer in a window neighborhood. FastPointTr[28] first voxelizes the point cloud and then applies the transformer in a local region. Given the substantial number of points in a scene's point cloud, applying the Transformer globally may seem impractical. Our key insight is that it's unnecessary to apply the Transformer to all points to capture long-distance context; applying the Transformer to key points is sufficient.

## 3 METHOD

Our goal is to explicitly model long-distance context across the entire scene with Transformer, rather than within local blocks. Our approach comprises two main parts, as illustrated in Figure 2(b): a point cloud backbone network for generating dense semantic predictions and a separate new branch that models long-distance contextual information and aggregates this information into the point cloud backbone. We will first review existing point cloud networks and then delve into the details of our design.

For the convenience of the discussion that follows, we will only use $x$ to denote the feature of a point without using a variable to denote $xyz$ unless otherwise stated in the text. But the $xyz$ of a point will be used in a $k$NN query process(including distance calculation) by default.

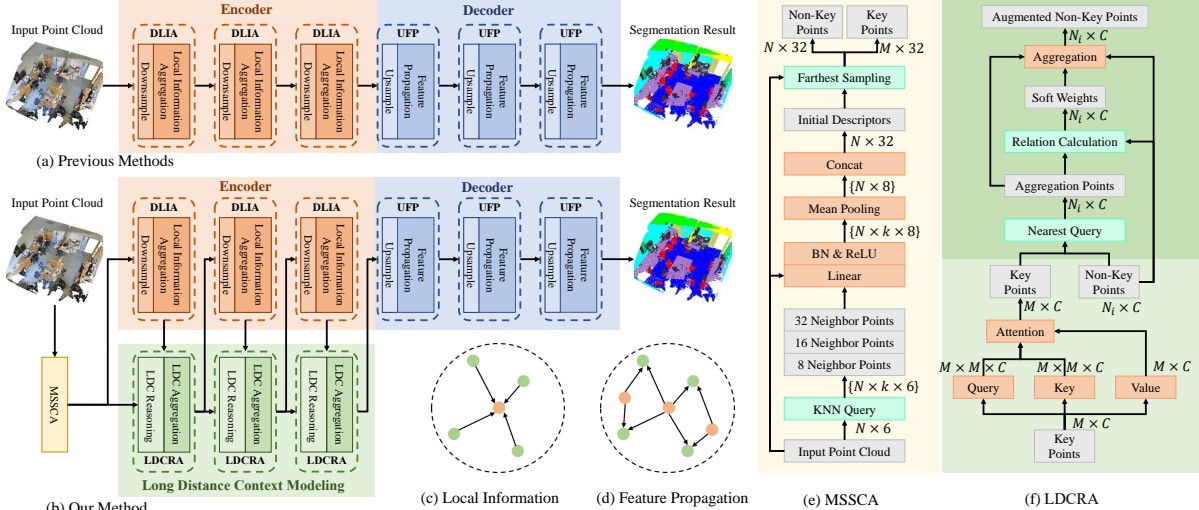

**Figure 2: Previous point-based semantic segmentation networks can be summarized in (a) and our method extends these methods by adding long-distance context modeling and aggregation Module in (b). (c) demonstrates the local information aggregation module, green points and orange points represent the points before downsampling and after downsampling respectively. (d) demonstrates the feature propagation module green points and orange points represent the points after upsampling and before upsampling respectively. (e) demonstrates the multi-scale spatial context aggregation(MSSCA) module and (f) demonstrates the long-distance context reasoning and aggregation(LDCRA) module.**

## 3.1 Review of Existing Point Cloud Networks

The existing point cloud networks are summarized in Figure 2(a). Generally, a network includes an encoder and decoder. The encoder has multiple downsampling and local information aggregation (DLIA) modules, while the decoder has upsampling and feature propagation (UFP) modules. Note that Figure 2(a) is a simplified case. Typically, there are 4-5 DLIAs and UFPs in the network, with a skip connection from DLIA to UFP, similar to UNet[32].

The DLIA module includes a downsampling module and a local information aggregation module. The downsampling module reduces point cloud resolution, similar to pooling layers in image processing. For example, RandLANet [22] uses random downsampling; KPConv [35] applies voxel downsampling; while PointNet [29], PointTr [52], and StratifiedTr [18] use farthest downsampling. The local information aggregation module aggregates point cloud features before and after downsampling. For each point after downsampling, it identifies the nearest $k$ points before downsampling and aggregates their features. For example, PointNet++ [30] uses max-pooling; KPConv [35] employs point convolution; and PointTr [52] utilizes a local self-attention mechanism for aggregation.

The UFP module includes an upsampling module and a feature propagation module. The upsampling module increases the point cloud resolution, usually achieved by preserving points after downsampling from different layers. For example, in a network with 4 DLIAs and 4 UFPs, the input point of the 1st UFP upsampling module is the output point of the 4th DLIA downsampling module, and the output is the output point of the downsampling module in the 3rd DLIA, and so forth. The feature propagation module propagates point cloud features before upsampling to the point cloud

after upsampling. Specifically, for each point $x_{au}$ after upsampling, finding the nearest $K$ points $\{x_{k,bu}\}$ before upsampling, and then propagate the features of those $K$ points by

$$x_{au} = \Theta\left(x_{au} + \sum_{k=1}^{K} w_k \cdot x_{k,bu}\right) \quad (1)$$

If the network has no skip connections, the initial value of $x_{au}$ is an all-0 vector. $\Theta$ is usually implemented with MLPs. $w_k = \frac{1/d_k}{\sum_{k=1}^{K} 1/d_k}$, where $d_k$ is the distance between $x_{au}$ and its $k$-th nearest neighbor $x_{k,bu}$.

## 3.2 Long-Distance Context Modeling

In this section, we will delve into the details of our design. First, a module named Multi-Scale Spatial Context Aggregation (MSSCA) is designed to extract key points and initiate descriptors. Then, to effectively model long-distance contextual information, a module named Long-Distance Context Reasoning and Aggregation (LD-CRA) is designed to capture global context and propagate information back to the original point cloud backbone.

*3.2.1 Multi-Scale Spatial Context Aggregation.* To ensure that the contextual scope encompasses the entire scenario, two key prerequisites must be met: 1) the key points must span every corner of the scene, and 2) each keypoint descriptor should be sensitive to localized information around the keypoint.

There are many keypoint extraction methods to choose from, including traditional methods[34, 53, 55] and deep-learning-based

methods[2, 36, 46]. We exclude deep-learning-based methods initially because they introduce too many additional network parameters and unnecessary memory footprint. Among traditional methods, random sampling loses significant information. Despite having the lowest time complexity, in practice, we observed that the model fails to converge. Other methods such as ISS[53], Harris3D[34], and Sift3D[55] are locally geometry-sensitive and do not guarantee the uniform distribution of keypoints. In contrast, farthest point sampling not only ensures the uniformity of sampling but also guarantees that keypoints cover the entire scene. Moreover, the farthest sampling method makes it easier for us to set the required number of keypoints.

Providing a robust initial descriptor for key points is essential for high-quality long-distance context modeling. The concept revolves around ensuring that each initial descriptor encapsulates neighborhood information at various spatial scales around the key point. This has the advantage of leveraging multi-scale information, enhancing the ability to recognize objects of different sizes within the scene, as mentioned in PatchFormer[49]. To achieve this goal, we introduce a simple module named the Multi-Scale Spatial Context Aggregation Module (MSSCA). This module queries neighbors of a key point at different scales and aggregates information from these neighbors to the key point.

The detail of the module is shown in Figure 2(e). For an input point cloud $P \in \mathbb{R}^{N \times 6}$, where the first three channels are $xyz$ coordinates and the last three channels are $rgb$, We first query multiple $k$-nearest neighbors of each point(specifically, $k = \{8, 16, 32\}$) and get $P_8 \in \mathbb{R}^{N \times 8 \times 6}, P_{16} \in \mathbb{R}^{N \times 16 \times 6}, P_{32} \in \mathbb{R}^{N \times 32 \times 6}$. Then we use a single MLP to lift their feature dimensions. After that, we use average pooling to aggregate its nearest neighbor information and concatenate this information with the original descriptor to obtain the initial descriptor of each point. Finally, we use the farthest sampling to sample $M$ key points. The key points $X_{key} \in \mathbb{R}^{M \times 32}$ are fed into LRDRA and the non-key points $X_{nkey} \in \mathbb{R}^{N \times 32}$ are fed into DLIA modules.

### 3.2.2 Long-Distance Context Reasoning and Aggregation.

In the original point cloud backbone, the input feature of the latter layer of encoders is derived from the outputs of the previous layer of the network. In the previous discussion, we observed that the features from the upper layer network encode only local information, which is not sufficient for semantic segmentation of large-scale scenes. Therefore, our idea is to perform an augmentation at each layer of the encoder of the original point cloud backbone network to adequately supplement long-distance contextual information into the original point cloud backbone network.

Specifically, each LDCRA module contains two parts: a Long-Distance Context Reasoning Module (LDCR) and a Long-Distance Context Aggregation Module (LDCA). Given keypoints and their descriptors, LDCR uses a standard self-attention mechanism to capture the relationship between two key points and model global contextual information. Given the global information carried by each key point, as well as the non-keypoint information, LDCA aggregates the global information from the key points to the non-keypoints.

**Long-Distance Context Reasoning.** Details of the long-distance context modeling are illustrated in Figure 2(f). For an input key point $X_{key}$, we use standard vector self-attention[52] as follows:

$$x_i = \sum_{m=1}^{M} W_w(W_q(x_i) - W_k(x_m) + P_r) \odot (W_v(x_m) + P_r) \quad (2)$$

where $x_i \in \mathbb{R}^C$ represent the descriptor of the i-th key point, $W_q, W_k, W_v$ represent $query, key$ and $value$ matrices respectively as defined in [38]. $W_w$ represents a weight layer that generates an attention vector. Following PointTr[52], we add a position encoding $P_r = \theta(p_i - p_m)$ to both the attention vector and the transformed features $W_v(x_m)$ to keep position information intact, where $p_i$ is the $xyz$ coordinate of the $i$-th key point.

**Long-Distance Context Aggregation.** The entire process is illustrated in Figure 2(f). A simple intuition is that adjacent points are more relevant. For each non-key point, we identify the nearest key point and incorporate the information carried by that key point to the non-key point. To further reduce noise and eliminate redundant information, we compute the difference between the two using a channel-wise subtraction operation. We then employ MLP and Softmax to generate a channel-wise soft weight. This soft weight is multiplied with the keypoint features and added to the non-keypoint features, resulting in the final augmented non-keypoint features. Specifically, given a key point $X_{key}^i \in \mathbb{R}^{M \times C}$ updated by self-attention at the $i$-th layer and a non-key point $X_{nkey}^i \in \mathbb{R}^{N_i \times C}$ from the $i$-th DLIA, the entire process can be formulated as:

$$x_{i,ne} = x_{i,ne} + \Theta(x_{i,ne} - x_{j,e}) \odot x_{j,e} \quad (3)$$

where $x_{i,ne}$ represents the $i$-th non-key point feature, $x_{j,e}$ represents the $j$-th key point feature. Also, $x_{j,e}$ is the of the point closest to $i$-th non-key point. $\Theta$ is implemented with MLPs but not the same as in Equation1.

## 3.3 Discussion

In this subsection, we discuss in detail the impact of each parameter on our approach and the design choices.

**Memory Footprint.** Here we focus on analyzing the additional memory footprint we introduced. Compared to applying the standard self-attention module globally to all points, since our self-attention mechanism is executed at key points and uses a vector attention mechanism, the total space complexity is reduced from $O(N^2C)$ to $O(M^2C)(M << N)$, where $N$ and $M$ are the numbers of origin point clouds and key points.

**Number of $M$.** Increasing $M$ theoretically allows for better capture of long-range context; however, the algorithm's space complexity scales with $M^2C$, leading to a tradeoff. Our experiments show some performance gains (Table 4), but considering the additional memories, we find this improvement less satisfying. Given that 256 keypoints already achieve a 3% mIoU boost compared to the original backbone network, we opt for the more memory-effective 256 keypoints.

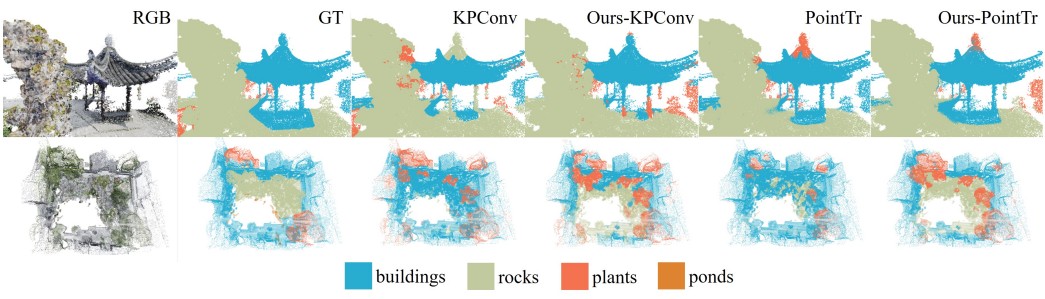

Figure 3: Visualization on the CHNRockery3D dataset.

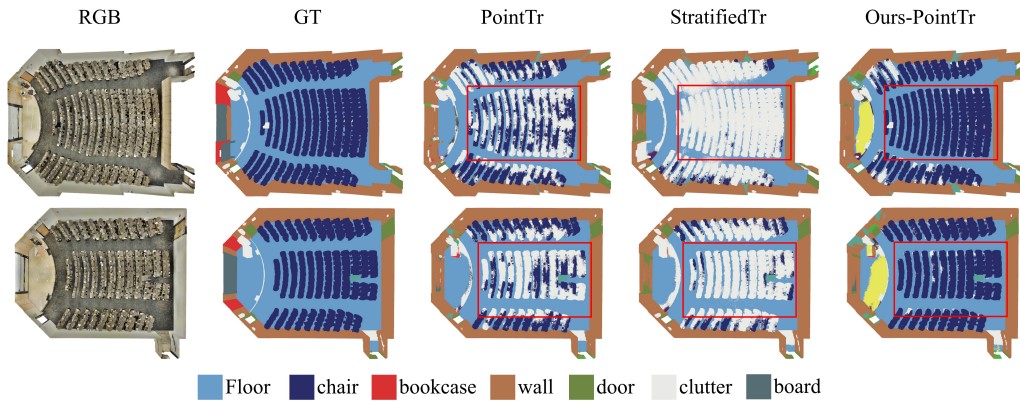

Figure 4: Results on 2 scenes in Area2 from S3DIS datasets. These two scenes contain large regions composed of a lot of small objects. Our method yields consistent segmentation results.

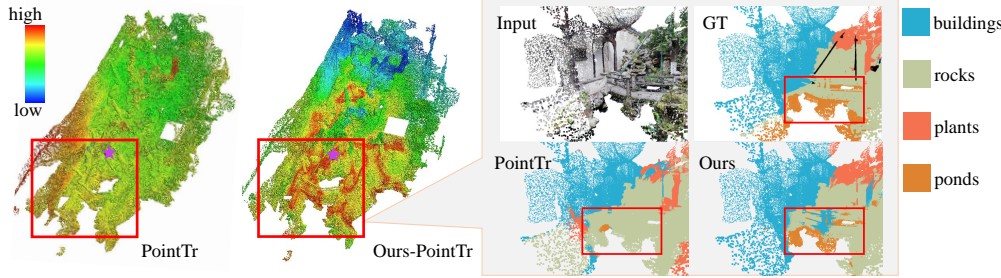

Figure 5: Visualization of ERF on ponds. Our method associates the pond to a larger region of the scene for inference.

## 4 EXPERIMENTS

We conducted experiments on six datasets including one unpublished dataset CHNRockery3D, as well as five public datasets: S3DIS [1], ScanNet[7], ScanNet200[33], Semantic3D[13], and SemanticKitti [3].

**CHNRockery3D Dataset.** Chinese rockery gardens, as material manifestations of gardening art, are an important tangible cultural heritage, while their design concepts and horticultural techniques also contribute to the intangible cultural heritage of China. On the one hand, systematic research and analysis of their digital assets is of great significance for the conservation and inheritance of heritage. On the other hand, Chinese rockery garden scenes are a very special kind of scene that has both the characteristics of

outdoor natural scenes (such as Semantic3d and SemanticKitti) and indoor artificial scenes (such as S3DIS and ScanNet). The objects in Chinese rockery gardens exhibit the characteristics of natural elements, such as vegetation and landscapes, but they are intentionally crafted and arranged by humans. The study of its features is informative and referential to the study of other scenarios. To explore the distinctive features of this special scene, we initially constructed the CHNRockery3D dataset and validated the effectiveness of our method on this dataset.

**Evaluation Protocol.** The CHNRockery3D dataset comprises seven classical Chinese rockery garden scenes. Each scene contains billions of points, covering an area exceeding 2000 square meters. Each point in the scan is assigned a semantic label from one of four

**Table 1: Comparison with baselines on the S3DIS dataset.**

| Method | Area-5 | | | 6-Fold | | |
|--------|--------|------|------|--------|------|------|
| | OA | mAcc | mIoU | OA | mAcc | mIoU |
| SPG[19] | 86.4 | 66.5 | 58.0 | 85.5 | 73.0 | 62.1 |
| SPT[31] | - | - | 68.9 | - | - | 76.0 |
| RandLANet[14] | 87.2 | 71.4 | 62.4 | 88.0 | 82.0 | 70.0 |
| StratifiedTr[18] | 91.5 | 78.1 | 72.0 | - | - | - |
| FastPointTr[28] | - | 77.3 | 70.1 | - | - | - |
| PointVector[9] | 91.0 | 78.1 | 72.3 | 91.9 | 86.1 | 78.4 |
| AFGCN[51] | 91.1 | 77.9 | 72.3 | 91.7 | 85.1 | 77.7 |
| PointTrV2[52] | 91.1 | 77.9 | 71.6 | - | - | 73.5 |
| PointTr[52] | 90.8 | 76.5 | 70.4 | 90.2 | 81.9 | 73.6 |
| Ours-PointTr | 91.2 | 78.2 | 71.8 | 91.8 | 83.4 | 75.4 |

categories: buildings, plants, rocks, and ponds. To assess methods on CHNRockery3D, we employ a 7-fold cross-validation approach. We adapted and modified several related methods from their official codebases to suit this dataset. For the other five public datasets, we follow a common evaluation protocol [14, 52]. Evaluation metrics include mean classwise intersection over union (mIoU), mean of classwise accuracy (mAcc), and overall pointwise accuracy (OA).

**Implementation Detail.** We mainly use PointTr[52] as our backbone because both methods are Transformer-based. In the original paper, PointTr is a 5-layer setup. To evaluate the effectiveness of our method, we use a 4-layer setup. To further show the adaptability of our method to existing methods, we also choose a more classical method, KPConv[35], and kept all the settings (number of layers, etc.) from the original paper. Unless otherwise mentioned, we use PointTr as the backbone. All experiments are conducted on a single RTX3090Ti GPU. The number of key points is set to 256 in most experiments.

## 4.1 Efficiency of Long-Distance Context Modeling

To verify the effectiveness of our long-distance context modeling, we discuss and illustrate the following experiments.

**Results on CHNRockery3D.** We compared our method with several existing open-source methods. Table 2 illustrates that our approach achieves a 3-4% improvement over the baseline method when using both KPConv and PointTr as the backbone. This suggests that our method exhibits some level of generalization within existing frameworks. Notably, our method attains the best results among these methods when utilizing PointTr as the backbone. It's worth highlighting that our method outperforms existing methods in both the buildings and rocks categories. This is particularly significant as buildings and rocks typically occupy substantial regions within a scene, emphasizing the importance of a larger contextual range for such cases.

The visualization results are presented in Figure 3. Notably, our method demonstrates improvements in the building and rock categories compared to the previous network. In the second row of Figure 3, it's evident that both KPConv and PointTr misclassify large areas of rocks as buildings, whereas our method accurately segments the rock area. However, we observed a decrease in our

method's performance on vegetation compared to other methods. This might be attributed to the relatively small size of vegetation compared to buildings and rocks. Consequently, a larger contextual information might introduce some noise.

Ponds constitute a special class of objects that share geometric similarities with rocks, but their distribution in the scene differs from that of rocks. Consequently, recognizing this object presents a considerable challenge. As indicated in Table 2, effectively segmenting this object is a daunting task for existing methods. However, our approach shows some progress in this category. The potential reason for this lies in the network's ability to model long-distance context for object analysis. This enables the network to identify the object's location in the scene and may even infer its function within the entire scene, providing further validation of the effectiveness of our method.

**Visualization of Effective Receptive Field.** To understand how our method identifies ponds, we visualize the effective receptive field (ERF), as suggested in [22], in Figure 5. It is evident that our method associates the pond with a larger region of the scene during inference. This implies that our method may implicitly learn relationships between objects. The visualization underscores the effectiveness of our approach in modeling long-distance contexts.

**Memory Footprint Comparison.** To demonstrate the memory efficiency of our approach, we experimented with different settings of PointTr, including a 4-layer setting, a 5-layer setting, and the use of more neighboring points for DLIA. Additionally, we conducted a comparison with StratifiedTr, considering its utilization of a larger window to capture longer contexts.

As shown in Table 3, the aggregation of information using more neighborhood points leads to a significant increase in memory. This is because, as the number of neighborhood points grows, the network must compute and store a larger attention map during the forward propagation process. However, despite this increase, the performance does not see significant improvement. Similarly, the addition of layers to PointTr results in a sharp rise in the number of parameters and memory usage. While this brings some performance gains, our approach achieves relatively greater improvements without introducing excessive memory footprint. Although StratifiedTr outperforms PointTr, it comes with the highest number of network parameters, and memory usage. It's important to note that StratifiedTr is configured with 4 layers. This experiment underscores the superiority of our method.

We also compare our method with existing approaches on various public datasets to demonstrate the potential of our methods.

**Results on S3DIS.** The segmentation results depicted in Figure 4 demonstrate the notably accurate performance of our method in a large area containing numerous chairs, underscoring the effectiveness of our approach for such scenes. Additionally, we present the results for Area-5 and the 6-fold cross-validation in Table 1. It is evident from the table that our method exhibits performance gains compared to the baseline used, and achieves comparable results to other recent methods.

**Results on Semantic3D.** We present the segmentation results on this dataset in Figure 6 and Table 5. Our method demonstrates a performance improvement over the baseline. Additionally, we observed an interesting phenomenon, depicted in Fig. 7. As the input scene range becomes larger (We simulate different sizes of

Table 2: Comparison with baselines on the CHNRockery3D dataset.

| Methods | OA | mAcc | mIoU | buildings | plant | rock | pond |
|---|---|---|---|---|---|---|---|
| PointNet[29] | 66.12 | 49.03 | 31.75 | 36.40 | 21.13 | 63.14 | 6.33 |
| PointNet++ [30] | 69.86 | 56.28 | 37.18 | 44.27 | 29.62 | 63.41 | 11.43 |
| DGCNN[41] | 70.29 | 48.68 | 34.18 | 40.85 | 24.73 | 66.30 | 4.85 |
| SPGraph[19] | 75.56 | 56.59 | 37.34 | 46.20 | 32.72 | 69.57 | 0.88 |
| RandLANet[14] | 79.93 | 57.82 | 41.75 | 58.26 | 33.80 | 74.77 | 0.15 |
| StratifiedTr[18] | 74.31 | 65.07 | 42.03 | 58.52 | 44.03 | 65.40 | 0.18 |
| KPConv[35] | 73.72 | 59.65 | 41.32 | 58.65 | 41.31 | 65.30 | 0.01 |
| Ours-KPConv | 79.90 | 60.07 | 44.76 | 59.71 | 38.72 | 74.15 | 6.46 |
| PointTr[52] | 76.39 | 58.32 | 40.22 | 57.11 | 31.66 | 72.06 | 0.05 |
| Ours-PointTr | 80.93 | 63.13 | 45.25 | 62.90 | 37.30 | 75.85 | 4.95 |

Table 3: Memory footprint compared with baselines on CHNRockery3D dataset. 4 means a 4-layer setting, and [8,16,16,16] means the number of neighbor points used in each DLIA. 'Window-based' means the number of neighbor points used in each DLIA depends on the window size and the distribution of points.

| Methods | Params. | Mem. | OA | mAcc | mIoU |
|---|---|---|---|---|---|
| PointTr(4,[8,16,16,16]) | 2.8M | 4.5G | 76.39 | 58.32 | 40.22 |
| PointTr(4,[8,32,32,32]) | 2.8M | 6.9G | 67.85 | 53.27 | 33.74 |
| PointTr(5,[8,16,16,16,16]) | 7.8M | 7.1G | 78.70 | 58.97 | 41.91 |
| StratifiedTr(4, window-based) | 8.0M | 7.2G | 74.31 | 65.07 | 42.03 |
| Ours-PointTr(4,[8,16,16,16]) | 3.3M | 4.6G | 80.93 | 63.13 | 45.25 |

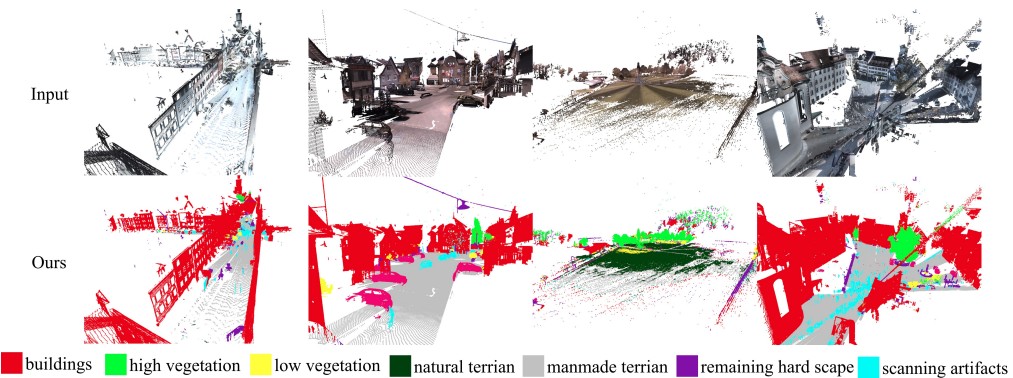

buildings   high vegetation   low vegetation   natural terrian   manmade terrian   remaining hard scape   scanning artifacts

Figure 6: Visualization on reduce-8 test set of Semantic3D dataset.

input scenes by cropping the scene from small to large ranges during the testing phase), our method achieves increasingly better performance, while the baseline method reaches a bottleneck at a certain level of scene range. This indicates that our method's capacity for modeling context can scale with the size of the scene, i.e. when the scene gets larger, our method can naturally extend the context-aware distance to a wider distance. This phenomenon validates the effectiveness of our long-distance context modeling. **Results on other datasets.** We also report results of our method on Scannet[7], Scannet200[33] and SemanticKitti[3] dataset in Table 5. In all three datasets, we made improvements from the baseline.

## 4.2 Ablation Study

We perform ablation experiments on CHNRockery3D to verify the effectiveness of each part of our method.

**Effectiveness of MSSCA.** Since this module is used to generate initial key points descriptors, removing this module means that only information from a single point is used when modeling long-distance dependencies. As shown in Table 4, the method performance drops a lot after removing the module, but it's still better than the 4-layer PointTr, which suggests that using a single-point feature can also be effective.

**Table 4: Ablation Study: Including three important designs - MSSCA, LDCRA, and SW; The influence of the number of keypoints $M$ and the location for modeling long-distance context (i.e. LDC loc.).**

| MSSCA | LDCRA | SW | OA | mIoU | $M$ | Mem. | mIoU | LDC loc. | OA | mIoU |
|:---:|:---:|:---:|:---:|:---:|:---:|:---:|:---:|:---:|:---:|:---:|
| ✓ | | | 78.29 | 41.01 | 128 | 4.3G | 44.75 | (-, -, -, ✓) | 79.56 | 43.48 |
| | ✓ | ✓ | 78.77 | 42.50 | 256 | 4.6G | 45.25 | (-, -, ✓, ✓) | 79.10 | 44.29 |
| ✓ | ✓ | | 78.33 | 44.11 | 512 | 5.9G | 45.60 | (-, ✓, ✓, ✓) | 75.55 | 41.05 |
| ✓ | ✓ | ✓ | 80.93 | 45.25 | 1024 | 15.6G | 45.56 | (✓, ✓, ✓, ✓) | 80.93 | 45.25 |

**Table 5: Results on other datasets.**

| Method | Scannet | | Scannet200 | | Semantic3D | | SemanticKitti | |
|:---|:---:|:---:|:---:|:---:|:---:|:---:|:---:|:---:|
| | OA | mIoU | OA | mIoU | OA | mIoU | OA | mIoU |
| MinkUNet[5] | - | 72.2 | - | 25.3 | - | - | - | 63.8 |
| SphereFormer [17] | - | - | - | - | - | - | - | 67.8 |
| PointTrV2 [44] | - | 75.4 | - | 30.2 | - | - | - | 70.3 |
| PointTr[52] | 89.4 | 70.6 | 81.9 | 29.7 | 90.0 | 64.5 | 91.9 | 61.7 |
| Ours-PointTr | 90.1 | 73.3 | 81.6 | 30.2 | 91.9 | 67.1 | 91.9 | 61.9 |

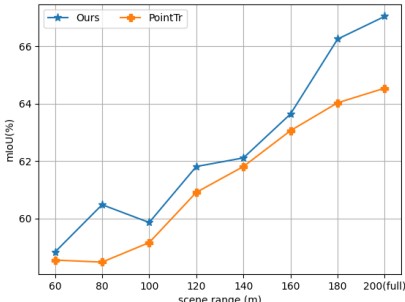

**Figure 7: Experiments on the ability of the method for context perception over long distances. It can be seen that our method can better extend the context-aware distance to a wider distance as the input scene range increases. For the size of the scene, we simply take the maximum value of the scene length and width.**

**Effectiveness of LDCRA.** We remove all LDCRA modules. In this case, the only difference between our method and PointTr is the input; PointTr uses the original point cloud as input, whereas removing LDCRA from our method is equivalent to using the initial descriptors extracted by MSSCA as input to PointTr. As shown in Table 4, we achieve a little bit of improvement compared to the origin PointTr. This demonstrates both the effectiveness of the MSSCA and the LDCRA.

**Effectiveness of Soft Weight(SW).** In our design, the soft weight is used to eliminate the redundancy between the critical point information and the non-critical point information, and after removing this weight, the performance is slightly degraded as can be seen in Table 4.

**Where to model long-distance context?** Our encoder has four layers, and in our design, we perform long-distance context modeling and fuse information with local contextual information at each layer. However, is it necessary to do so? For this purpose, we performed ablation experiments. As shown in Table 4 (i.e. LDC loc.), doing it at every layer is required to fully utilize the capabilities of our approach.

**Impacts of the number of the keypoints.** The impacts on $M$ is shown in Table 4. As the number of points increases, the memory overhead increases dramatically, and the performance gain it provides is relatively modest. 256 achieves a performance-efficiency tradeoff.

## 4.3 Limitation

Although our approach models contextual information over longer distances with only a slight increase in memory overhead, the extensive use of $k$NN searches, as well as farthest point sampling algorithms throughout the network's inference, results in a significant time overhead. One avenue for future improvement is the serialization of the point cloud. Point cloud serialization [21, 40, 43] is a recent technique that has emerged, showing promise in greatly enhancing overall inference speed by appropriately ordering the point cloud so that nearest neighbor points are adjacent to each other in memory. This approach leads to an almost $O(1)$ complexity compared to the $O(N^2)$ complexity required for $k$NN and farthest point sampling.

## 5 CONCLUSION

This paper introduces a novel approach to explicitly model long-distance contextual information throughout the entire scene, effectively addressing the ambiguity challenges inherent in large-scale scenes. Our method achieves this with minimal memory footprint. Experimental results demonstrate the effectiveness of our approach across a range of scene cases, particularly excelling in challenging scenarios involving irregular large-sized objects. In future work, further exploration of the definition and challenges associated with large-scale scenarios is needed.

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
