# OpenReview forum: "LDCNet: Long-Distance Context Modeling for Large-Scale 3D Point Cloud Scene Semantic Segmentation"
_acmmm.org/ACMMM/2024/Conference — MM2024 Poster_

### Official Review · Reviewer_S1MK · 2024-05-21

**Rating:** 4
**Confidence:** 3

**Summary:**

The paper tackles the point cloud semantic segmentation task and focuses on the long-distance context modeling problem. The proposed method is intuitive and effective, performing well on several representative datasets.

**Strengths:**

The stated problem in the task is both authentic and crucial. The proposed method directly and convincingly addresses the problem. The writing is clear and easy to follow, and the experiments are abundant, accompanied by in-depth analysis.

**Limitations:**

1. Time Consumption Analysis: Although L907 mentions significant time overhead, the paper lacks related statistics, which is unacceptable.
2. CHNRockery3D Dataset Availability: Will the CHNRockery3D dataset be made public? It would serve as a valuable testbed for the community.
3. Exploring Key Point Extraction Methods: Beyond fps, it's crucial to explore various sampling methods and understand their impact on the proposed long-distance context modeling approach.

**Suitability:**

2

---

### Official Review · Reviewer_4rez · 2024-05-25

**Rating:** 2
**Confidence:** 3

**Summary:**

This paper introduces LDCNet, a method for large-scale 3D point cloud scene semantic segmentation, which effectively models long-distance contextual information by using key points and local descriptors to understand the overall scene layout. Several experiments are conducted, demonstrating this approach can effectively mitigate ambiguities and has good generalization for typical scenarios

**Strengths:**

1) The paper introduces a new branch that models long-distance contextual information and aggregates this information into the point cloud backbone, enhancing semantic segmentation accuracy for large-scale 3D point cloud scenes.

2) The proposed method explicitly model long-distance context across the entire scene with Transformer, rather than within local blocks, achieving better results while saving computational resources.

3) The method description of the article is easy to understand, and there are much visualization to help understand.

**Limitations:**

1) Point sampling for keypoint is important in the MSSCA module, and the comparison of different sampling methods are discussed in the method part. However, there is no ablation experiment on the influence of different sampling methods in the experiment part to support the discussion in the method part.

2) About the LDCRA module, since the data will be downsampled in each stage of the encoder, will the keypoints also be downsampled in each stage of the LDCRA module? In addition, the right part of table 4 shows that the relationship between the number of LDCRA modules and the performance of the whole model does not increase linearly. There is a lack of theoretical analysis for this phenomenon.

3) Although the method proposed in this paper has some improvements over the chosen backbones, the experimental results on some datasets shown in the experiment part have no obvious advantages compared with SOTA. More experiments on the performance improvement of this method under different backbone are needed.

4) The experiment part only introduces the dataset CHNRockery3D in detail, and lacks a brief description of the content of several other datasets.

**Suitability:**

2

---

### Official Review · Reviewer_wZXL · 2024-06-04

**Rating:** 4
**Confidence:** 3

**Summary:**

This paper deals with the problem of 3D point cloud semantic segmentation in large-scale scenes. A Long-Distance Context modeling method, named LDCNet, is proposed. The method first extracts key points and encodes contextual information using Transformer to facilitate semantic segmentation. Extensive experiments demonstrate the effectiveness of the proposed method. The manuscript is well-written and structured.

**Strengths:**

The task of large-scale point cloud scene semantic segmentation is of vital importance to autonomous driving. This work tries to improve the performance of the task from the perceptive of long-distance contextual information encoding to under the scene layout. The mechanism effectively mitigates the issue of ambiguity during scene feature encoding. Experiments on several dataset show the strength of the work.

**Limitations:**

I believe the manuscript can be further improved regarding the following concerns:
1. The basic idea of the work is that the extracted key points can be utilized to model or encode the entire scene. Such an idea is not new. Please refer to PointNet++ used to encode the geometric information of a 3D object, while the work is an extension to the large-scale 3D scene.
2. The definition of the loss function associated with the training of the network is missing.
3. The depiction of Figure(e) should be corrected, regarding the feature extraction using 8, 16, 32 neighbors followed by MPL.

**Suitability:**

2

---

### Official Review · Reviewer_3Mrw · 2024-06-06

**Rating:** 4
**Confidence:** 4

**Summary:**

This paper introduces a Long-Distance Context Modeling Network (LDCNet) for Large-Scale 3D Point Cloud Scene Semantic Segmentation. LDCNet aims to resolve ambiguities arising from locally high inter-class similarity by extracting key points along with a local descriptor.Ths work is highly followed by PointTr. The performance was evaluated on six datasets. The idea and topic of this manuscript are interesting and have potential significance

**Strengths:**

The presentation of the paper is clear. The experiments are conducted in a correct manner, and both qualitative and quantitative results are compared with an ablation study.

**Limitations:**

1. It is mentioned in line 109 that the transformer model increases the computational footprint, but this work heavily follows the transformer-based PointTr model, which is a significant contradiction. It is expected to explore something new beyond transformer-based models.
2. The contribution of this work is claimed to be the extraction of key points using conventional descriptors rather than DL-based models to reduce computational complexity. However, no theoretical support or discussion is provided for key point extraction.
3. Key point-based methods often focus on local geometric features, so how the model maps or learns global features in such complex, cluttered environments need to be discussed.
4. Another factor is the selection of key points, which is limited to 256 in this paper. How the model selects dynamic key points in large-scale contexts, especially given that point clouds can vary greatly in scale and point density, needs to be considered and discussed further.

**Suitability:**

3

---

### Meta-Review · Area_Chair_Gtp4 · 2024-06-29

**Recommendation:** Accept (Poster)
**Confidence:** 3

**Metareview:**

Reasonable quality but lacks convincing performance.